# Assessment of Common Risk Factors of Non-Communicable Diseases (NCDs) and Periodontal Disease in Indian Adults: An Analytical Cross-Sectional Study

**DOI:** 10.3390/mps5020022

**Published:** 2022-03-03

**Authors:** Lakshmi Puzhankara, Chandrashekar Janakiram

**Affiliations:** 1Department of Periodontology, Manipal College of Dental Sciences, Manipal Academy of Higher Education, Manipal 576104, India; 2Department of Public Health Dentistry, Amrita School of Dentistry, Amrita Vishwa Vidyapeetham, Ernakulam 682041, India

**Keywords:** clustering, common risk factor approach, non-communicable diseases, periodontal disease, risk factors

## Abstract

Risk factors that predispose individuals towards major non-communicable diseases (NCDs) and periodontal disease (PD) often co-occur in the same individual. The common risk factor approach (CRFA) for controlling the risk factors associated with NCDs and PD ensures that modifying a few risk factors has an incredible impact on regulating many chronic conditions. To apply CRFA to NCDs and PD, it is essential to quantify the common risk factors of these conditions. The proposed hospital-based analytical cross-sectional study aims to assess the proportion overlap of risk factors that are common or shared between NCDs (cardiovascular diseases (CVD) and diabetes mellitus (DM) type 2) and PD. The risk factors for PD and NCDs will be estimated in subjects aged 18 years and above, diagnosed with NCDs (DM type 2, CVD) or PD. This will be a non-directional study. The dependent variables analyzed will be PD and NCDs (DM type 2, CVD). The explanatory variables that are assessed will be: age, gender, address, occupation, access and affordability of care, familial pattern, family size, insurance, socioeconomic status, obesity, tobacco usage, physical activity, alcohol consumption, food frequency, stress, and oral hygiene. The research is expected to provide data which will aid in the development of advocacy initiatives to implement CRFA for PD and NCDs.

## 1. Introduction

Non-communicable diseases (NCDs) are chronic diseases that are not directly transmissible from one person to another. They depict the true meaning of the word insidious, as they result in over 71% of all deaths globally and often remain undetected until the condition has reached a near-fatal stage especially in low and middle income countries [1]. The major NCDs include cardiovascular diseases, cancers, chronic respiratory diseases and diabetes [1]. Of the 10 leading causes of deaths in 2019, 7 were NCDs and these NCDs accounted for 44% of all deaths globally. All NCDs, taken together, accounted for 74% of deaths globally in 2019 [2]. Tobacco use, physical inactivity, unhealthy diet and the harmful use of alcohol are modifiable risk factors that can lead to an increase in the risk of NCDs. Elevated blood pressure, being overweight/obesity, hyperglycaemia and hyperlipidaemia are the metabolic risk factors that can contribute towards an increase in the risk of NCDs [3]. Stress and social determinants of health such as a person’s economic and social conditions can also increase the risk of NCDs. The burden of NCDs on national economies is tremendous, with an estimated loss of 9 billion international dollars in national income in India in 2005 due to death or inability to work as a result of heart disease, stroke or diabetes [4]. The economic burden of NCDs in India is in the range of 5–10% of gross domestic product (GDP) [5].

Periodontitis, as an inflammatory, non-communicable disease with a multifactorial aetiology, is an extremely prevalent condition. Amongst the adult population worldwide, more than 10% are affected by severe periodontitis and the condition may even be more prevalent than cardiovascular diseases. Periodontal disease has an estimated 7.4% prevalence and thus is estimated to be the 11th most prevalent disease globally with around 538 million people affected [6]. In India, the overall prevalence of periodontal disease was estimated to be 51% and the prevalence of gingivitis was 46.6% [7]. Periodontal diseases are assessed as being responsible for 3.5 million adjusted years’ worth of living with a disability [3]. Periodontitis reduces quality of life, is an encumbrance to the healthcare economy [6,8] and shares numerous risk factors such as smoking, obesity, nutrition, low socioeconomic status and stress with other non-communicable diseases (NCDs) such as diabetes, cancer, cardiovascular and cerebrovascular diseases [9,10,11,12]. A total of 54 billion USD/year is the estimated global cost of lost productivity due to periodontitis and in the year 2010, 440 billion USD (direct and indirect costs) was the total economic impact incurred due to direct and indirect factors associated with periodontal disease. The National Health Service of the UK spent over 2.8 billion GBP in 2017 towards the management of periodontal diseases [13].

The co-occurrence of several characteristics in one individual is termed as clustering. Clustering of risk factors has been observed in relation to NCDs. [14] Behavioral and demographic characteristics that predispose individuals towards major chronic diseases frequently cluster in the same individuals. Unhealthy diet, sedentary lifestyle and deleterious life habits are often seen grouped together in people [15].

The common risk factor approach (CRFA) for tackling the common risk factors associated with NCDs has been cresting for a while in the world of healthcare. CRFA is based on the scientific concept that if disease A is caused by C and disease B is caused by C, then the reduction of C should facilitate the reduction of both A and B, that is, controlling a small number of risk factors can have a tremendous impact on controlling a large number of chronic conditions [16]. The clustering of risk factors in individuals enables effective reduction of conditions initiated by these risk factors through the implementation of CRFA. Several programs have been initiated in which CRFA has been implemented to achieve better general health [17,18]. A few advocacy initiatives focusing on CRFA in relation to oral health and general health have attempted to forge the relation between NCDs and oral health through the commonality of risk factors and thus initiate an active reduction of conditions through CRFA [19,20,21,22]. The World Health Organization (WHO) has recently adopted a global strategy for oral health with the aim of improving oral health in conjunction with overall health [23].

Integration of the prevention of periodontal disease along with public health approaches for systemic health, particularly with non-communicable disease prevention activities focusing on common risk factors, can pave the way towards better overall health with a concomitant reduction in health care costs. Consequently, the model of the common risk factor approach (CRFA) has been hypothesized as a public health approach to prevent periodontal disease.

The limited availability of numerical data on the influence of various risk factors on NCDs and on periodontal disease makes it difficult to bring about a fool-proof plan to create a convincing argument favoring the common risk factor approach. This research is designed to overcome the deficiency in the literature in relation to the quantification of common risk factors between periodontal disease and NCDs. Policies are based on evidence and evidence is based on statistics. The quantification of the commonality of risk factors between NCDs and periodontal disease could result in the development of advocacy initiatives which would help in the reduction of risk factors associated with periodontal disease and NCDs.

This proposed study is based on the rationale that periodontal disease shares a wide range of risk factors and risk indicators such as tobacco use and alcohol, dietary factors, stress, plaque microbiota, oral hygiene and social determinants of health (SDH) with other systemic disease conditions, particularly the other major NCDs such as cardiovascular disease, chronic respiratory disease, cancer and diabetes. Quantification of these risk factors could enable the implementation of CRFA for the integration of periodontal disease prevention into public health approaches for NCD prevention and control activities.

### 1.1. Research Question

What is the proportion overlap of the risk factors that are common or shared between non-communicable diseases (cardiovascular diseases and diabetes mellitus type 2) and periodontal diseases?

### 1.2. Hypothesis

A significant proportion of risk factors are more likely to be shared between periodontal disease and other major non-communicable diseases (NCDs—diabetes mellitus type 2 and cardiovascular disease) in adults.

### 1.3. Objective

The proposed research aims to bridge the gap in knowledge regarding the commonality of risk factors between periodontal disease and NCDs and provide the data required for a CRFA encompassing periodontal disease and NCDs. While it is known that there are several risk factors common to both NCDs and periodontal disease, there is a lack of understanding regarding the extent of the overlap of the risk factors. The risk factors that predispose individuals towards periodontal disease and NCDs have not been quantified. 

The objective of this research is to assess the proportion overlap of the risk factors that are common or shared between non-communicable diseases (cardiovascular diseases and diabetes mellitus type 2) and periodontal diseases.

## 2. Experimental Design

### 2.1. Study Design

The proposed analytical cross-sectional study will focus on the assessment of the proportion overlap of the risk factors that are common or shared between non-communicable diseases (cardiovascular diseases and diabetes mellitus type 2) and periodontal diseases.

We will estimate the risk factors for periodontal disease and risk factors for non-communicable disease. The risk factors and outcomes are already present. It will be a non-directional study (Figure 1).

### 2.2. Study Setting

This will be a hospital-based study conducted in tertiary care medical college teaching hospitals (Kasturba Medical College and Hospital (KMC) and Amrita Institute of Medical Sciences and research center (AIMS)) and dental hospitals (Manipal College of Dental Sciences (MCODS), Manipal and Amrita School of Dentistry, Kochi) in South India.

### 2.3. Participants

The subjects will be selected based on the presence or absence of the conditions (NCDs and/or periodontal disease) and divided into three groups as follows: Patients with either one or more NCDs with periodontal disease will belong to Group A. Patients with either one or more NCD and no periodontal disease and patients with periodontal disease and no NCDs will be in Groups B and C, respectively.

Subjects with no NCD or periodontal disease, subjects not willing to participate in the study and moribund patients will be excluded from participating in the study.

Patients diagnosed with non-communicable diseases (diabetes mellitus type 2 and cardiovascular diseases) who are adults aged 18 and above will be selected. The disease/condition will be defined as follows: I.Diabetes mellitus type 2 is diagnosed if a patient has any two of the following findings [24]:The presence of symptoms of diabetes such as increased thirst, increased urination and the random plasma glucose test shows a glucose level of 200 milligrams per deciliter (mg/dL).The fasting blood sugar level is equal to or greater than 126 mg/dL.A 2-h oral glucose tolerance test (OGTT) is ≥200 mg/dL.The hemoglobin A1c (HbA1c) is 6.5% or higher.
II.Cardiovascular disease is defined as the name for the group of disorders of the heart and blood vessels [25], and includes:(a)Hypertension (high blood pressure) diagnosed by the following findings (Table 1): [26].(b)Coronary heart disease (heart attack)/myocardial infarction (MI) diagnosed by abnormalities detected in an electrocardiogram (ECG), echocardiogram, exercise stress test, a nuclear stress test, cardiac catheterization, an angiogram, a cardiac CT scan or CT coronary angiogram [27].(c)Cerebrovascular disease (stroke) is diagnosed if there are abnormalities detected in tests such as [28,29] physical exams—stroke scale [30], blood tests, computerized tomography (CT) scan, magnetic resonance imaging (MRI), carotid ultrasound, cerebral angiogram or echocardiogram.(d)Peripheral vascular disease diagnosis occurs through the detection of abnormalities during physical examination, blood tests, in the ankle–brachial index (ABI), in an ultrasound or angiography [31,32].

III.Periodontal disease is defined as an inflammatory process affecting the supporting structures of the teeth resulting in pocket formation, recession or both and is diagnosed with a periodontal screening and recording (PSR) score of three in at least one sextant [33].

### 2.4. Participant Recruitment

The inclusion criteria will include the following characteristics:Patient will have at least one or more of the selected NCDs or periodontal disease as stated earlier.Patients should not have any comorbidities such as infectious diseases or other diseases other than non-communicable diseases taken into consideration.

The following subjects will be excluded:Subjects with no NCD or periodontal disease;Subjects not willing to participate in the study;Moribund patients.

Patients reporting to the Outpatient Department of Periodontology at Manipal College of Dental Sciences, Manipal and Amrita School of Dentistry, Kochi will be screened. If the patient fulfils the inclusion criteria, informed consent will be obtained from the patient. The patient will be allotted to Group A if the patient has any of the NCDs and has periodontal disease and to Group B if the patient has NCDs but no periodontal disease. The patient will be allotted to Group C if the patient has only periodontal disease with no NCDs. The attendees of the patients will also be screened to verify if they satisfy the inclusion criteria. If they satisfy the inclusion criteria, informed consent will be obtained from the participants and they will be allotted to the groups as considered appropriate. Furthermore, after consultation with the departments of endocrinology, cardiology and neurology of the medical college hospitals, patients reporting to these departments will also be screened for the criteria for inclusion in the study. After obtaining informed consent, based on the presence or absence of periodontal disease, these patients will be allotted to either Group A or Group B. The participant selection decision tree is given in Figure 2.

### 2.5. Outcomes

The primary outcome of this study will be an assessment of the proportion of risk factors more likely to be shared between periodontal disease and other major non-communicable diseases (NCDs—diabetes mellitus type 2 and cardiovascular disease) in adults.

The following will be analyzed:Dependent variables;Periodontal disease and non-communicable diseases (diabetes mellitus type 2, cardiovascular disease).

Explanatory variables:➢Age;➢Gender;➢Address;➢Occupation; ➢Access to care; ➢Affordability of care; ➢Familial pattern;➢Family size; ➢Insurance;➢Socioeconomic status (modified Kuppuswamy criteria) [34];➢Obesity–body mass index (BMI) is very commonly used to classify being overweight and obesity in adults. Body mass index (BMI) is measured as weight in kilograms/square of height in meters (Kg/m^2^) [35];➢Tobacco usage: yes/no;➢If yes: type/duration/frequency/quantity;➢A physical activity measurement tool will be measured by the Physical Activity Assessment Tool devised by American Family Physician [36];➢Alcohol consumption will be assessed by an alcohol consumption screening AUDIT questionnaire in adults introduced by WHO [37];➢Food frequency questionnaire [38];➢Stress can be assessed using the standard stress assessment questionnaire [39];➢Oral hygiene; ➢Simplified Oral Hygiene Index [40];➢Plaque index (Sillness and Loe) [41].

## 3. Procedure

### 3.1. Data Sources/Measurement

#### 3.1.1. Tools Used

Two instruments will be devised for this study: (a)Data collection forms for the clinical and laboratory investigations data and recording of anthropometric data;(b)Validated structured questionnaire forms for enumerating the risk factors of each of the NCDs.

A data collection form that determines the eligibility of the participant for inclusion in the study will be utilized. 

The data collection forms for the clinical data will include details on medical and dental history, dental visits in the past, anthropometric data and the results of laboratory investigations conducted. The form will also include the diagnosis and executed treatments during the most recent visit as well as the treatment strategy.

The initial section of the questionnaire will comprise the demographic characteristics of the study participant and the remainder will be related to the risk factors of the non-communicable disease in question.

Data collection will be through:

Patient interview;

Data from their medical records;

Oral examination;

Pathological reports or assessment if available.

#### 3.1.2. Data Collection

A data collection form that determines the eligibility of the participant for inclusion in the study will be used to determine the inclusion of patients in the study (Appendix A). The assessment of the risk factors of the NCDs and oral diseases (dental caries and periodontal disease) will be carried out by two methods: patient interview and data from their medical records. 

A validated structured questionnaire for enumerating the risk factors will be devised for the interview and the questionnaire will consist of two sections. The first section will comprise the demographic characteristics of the study participant (Appendix A) and the second part will have the details of the various risk factors of the non-communicable diseases including periodontal disease (Appendix A). The interview will be carried out by the Principal Investigator or Co-PI who will be trained on the conduct of the interview. The patient will be interviewed for approximately 15–20 min, when all the relevant questions regarding their exposure to the stipulated risk factors will be assessed. 

Clinical examinations will be conducted to record the anthropometric data (Appendix A) and the details on CVDs (Appendix A) and periodontal disease (Appendix A). The results of conducted laboratory investigations will be recorded. The patients’ medical records will be accessed to obtain missing information, validate the details provided by the patient and obtain the values of blood pressure, lipid and blood glucose levels (Appendix A). These details will be recorded in the data collection forms for the clinical and laboratory investigation data. The form will also include the diagnosis and executed treatments during the most recent visit as well as the treatment strategy (Appendix A). A clinical oral examination will be carried out to assess the periodontal status of the patient using a periodontal probe by the Principal Investigator (PI) or Co-PI. The PI and Co-PI will be calibrated prior to the initiation of the study. For calibration, the examiners will first have an alignment session, where examiners will review in detail the clinical parameters included in the study protocol. The second day, the assessment session, will be when examiners measure patients with the characteristics of the study and scores will be evaluated for both intra- and inter-reliability [42]. The Oral Hygiene Index (OHI) and the Periodontal Screening and Recording Index will be recorded in the data collection forms for the clinical data. The data collection forms are given in Appendix A.

### 3.2. Bias

The investigators will be calibrated to reduce the risk of bias in the data collection. Furthermore, only those subjects with a recent diagnosis of NCDs, that is, within 3 months of diagnosis, will be selected to reduce the recall bias. Assessment of the risk factors from MRD records as well as physician interviews to reconfirm risk factors will be conducted to reduce the risk of bias. 

### 3.3. Sample Size

Assuming a prevalence of 40% common risk factors among periodontal disease and any of the non-communicable diseases, with 80% power and 5% alpha error, the minimum sample size in each group was estimated to be 164 per disease group. To account for any missing data, the final sample size per group was increased to 200. Hence, a final sample size of 600 subjects will be selected. A split up of the sample size is given in Figure 3.

### 3.4. Statistical Methods

Data analyses will be conducted using the R Package. The data obtained will be tabulated and analyzed using descriptive statistics and risk factors will be expressed as percentages. Bivariate analysis using Cohen’s kappa will be conducted to assess the degree of agreement for each risk factor between the periodontal diseases and each of the NCDs. A 2 × 2 table will be obtained to calculate the degree of agreement, which will also be expressed in percentages. Further quantification of attributable risks that are common to both PD and each NCD will be conducted. Next, Polynomial regression will be used to predict or adjust the risk factors that are common to both conditions.

### 3.5. Data Monitoring

The compliance of the data to the protocol will be ascertained along with the accuracy of source documents. The standard operating procedures will be followed and the monitors will verify that the research is conducted and reported in accordance with the protocol.

### 3.6. Ethical Considerations

The study protocol was submitted to the Institutional Review Board of the participating institution. After scientific approval, the protocol was submitted for ethical approval to the Institutional Ethics Committee (IEC) of Kasturba Medical College (KMC) and Hospital, Manipal.

This research will be conducted following the principles of the Declaration of Helsinki and local guidelines (Indian Council of Medical Research). An Annual Progress Report will be submitted by the researchers to the Institutional Review Board (IRB), KMC. At the end of the study, a closure report will also be submitted. Only authorized persons will be allowed access to final data sets.

### 3.7. Protocol Amendments

Any modifications to the protocol which may impact the conduct of the study or its outcomes will be reported to the IRB for approval. 

### 3.8. Consent to Participate in the Study

Participants of the study will receive verbal as well as written information regarding the research from one of the trained investigators. The participants will be given a detailed explanation regarding the research and its consequences including the implications and limitations of the protocol as well as the risks involved in taking part in the trial. It will also be made clear to the participants that they are free to withdraw from the study at any time for any reason without any deleterious consequences. The participants will be allowed to deliberate on the information they receive and to question the researcher in order to clarify their doubts regarding the research and its effects. Information sheets and consent forms in the spoken language of the participants will be provided for all subjects involved in the study.

Once a participant has arrived at a decision after due consideration of the information obtained, written informed consent will be obtained from the consenting participant. The dated signatures of the participant and the person who presented the information and obtained the informed consent will be affixed to the informed consent form. The original signed form will be preserved at the research site and the participant will be given a copy of the form. 

### 3.9. Confidentiality

The anonymity of the participants will be maintained, and a unique study-specific participant ID number will be used to identify each participant on all study documents or any electronic databases. The documents will be made accessible only to the staff involved in the research and authorized personnel. 

## 4. Expected Results

The quantification of risk factors that are common for periodontal disease and other major non-communicable diseases will help in providing a point of reference from which to measure the shifting patterns of disease, deliver data for policy development and guarantee that intervention activities at population levels are appraised suitably for effectiveness. Presently, only the attributable risk of smoking to periodontal disease has been sufficiently quantified. Without information about other main risk factors that are modifiable, explanations of the integration of periodontal disease with other non-communicable diseases for periodontal disease prevention through the common risk factor approach are not as compelling as they could be. This research would hence provide information that would help integrate the concept of risk factor reduction for the management of periodontal disease and NCDs.

## Figures and Tables

**Figure 1 mps-05-00022-f001:**
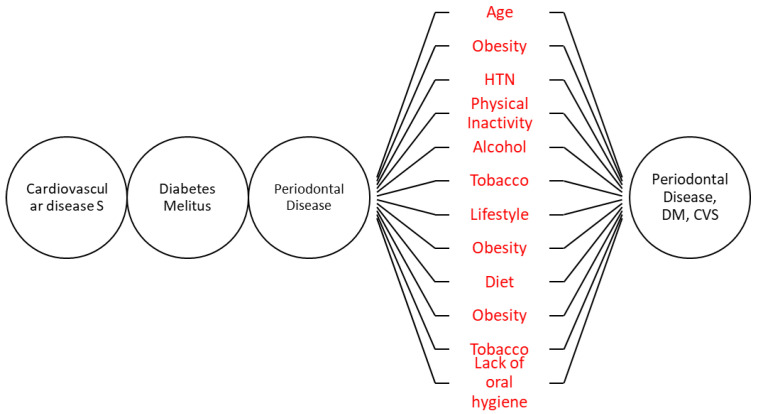
Study design.

**Figure 2 mps-05-00022-f002:**
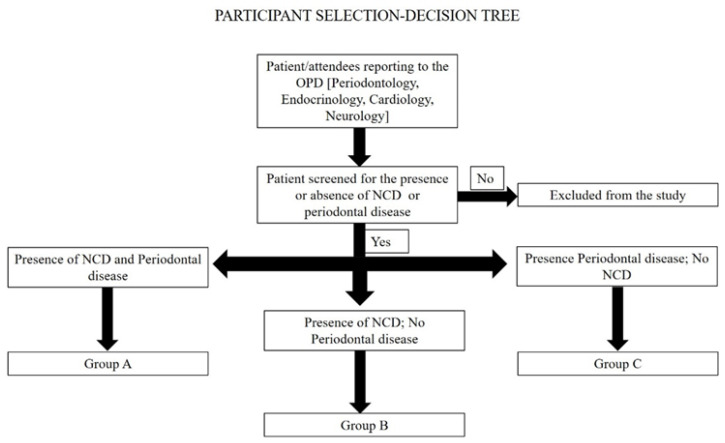
Participant selection–decision tree.

**Figure 3 mps-05-00022-f003:**
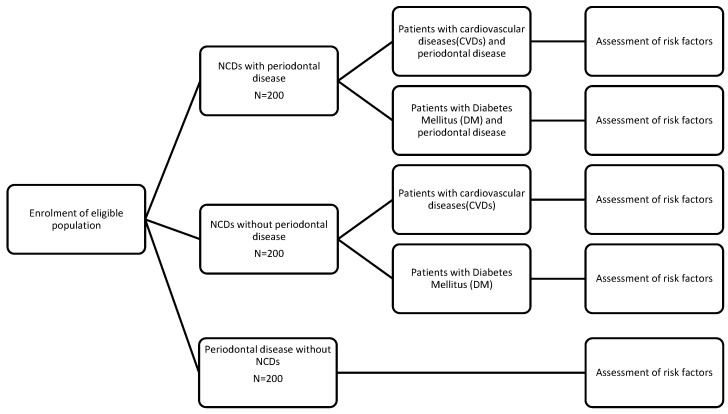
Split up of sample size.

**Table 1 mps-05-00022-t001:** Diagnosis of hypertension.

Blood Pressure Classification	SBP	DBP
	mmHg	mmHg
Normal	<120	<80
Prehypertension	120–139	80–89
Stage 1 Hypertension	140–159	90–99
Stage 2 Hypertension	≥160	≥100

## Data Availability

Data sharing is not applicable to this article.

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
