# Peer review of "Assessment of Common Risk Factors of Non-Communicable Diseases (NCDs) and Periodontal Disease in Indian Adults: An Analytical Cross-Sectional Study"

_mps, 2022, doi:10.3390/mps5020022_

Round 1
Reviewer 1 Report
The authors of the manuscript “Assessment of common risk factors of Non-communicable Diseases (NCDs) and Periodontal disease in Indian adults: An analytical cross-sectional study” aim to analyze (partly) common risk factors of both NCDs and PD.
Introduction is well written and provides an overview with outlining significance and aim of the study. Figures are sufficient.
Please tell more about group allocation and anonymization process.
How is oral and medical examination performed by PI and co-PI at multiple centers?
References are not in coherent style with missing data.
English language does need minor corrections.
Author Response
"Please see the attachment."

Reviewer 2 Report
Dear authors,
Thank you for submitting your valuable work to the journal. The threat to global healthcare caused by non-communicable disease is significant and more effort should be put into the control of modifiable risk factors than would lead to a decrease of their chance of occurance. Thus I believe the topic of the paper is intersting and that the link to periodontal disease is of outmost importance to dental specialists. However, there are some comments that I would make on the article, in order to improve its scientific accuracy:
- Please include indexing number for the scientific and ethical approval of the study
- Please state Exclusion criteria for participants
- Diagnosis of peridontal disease is based on an older algorithm. Please also reffer to the 2018 protocol of the European Federation of Peridontology and American Academy of Periodontology
We look forward to receiving the revised version of your manuscript.
Kind regards!
Round 2
Reviewer 1 Report
I would like to thank the authors for the revisions. Article is now acceptable.